# Compressed Prostate Cancer Cells Decrease Osteoclast Activity While Enhancing Osteoblast Activity In Vitro

**DOI:** 10.3390/ijms24010759

**Published:** 2023-01-01

**Authors:** Victor J. B. van Santen, Behrouz Zandieh Doulabi, Cornelis M. Semeins, Jolanda M. A. Hogervorst, Cornelia Bratengeier, Astrid D. Bakker

**Affiliations:** 1Department of Oral Cell Biology, Academic Centre for Dentistry Amsterdam (ACTA), University of Amsterdam and Vrije Universiteit Amsterdam, Amsterdam Movement Sciences, 1081 LA Amsterdam, The Netherlands; 2Division of Cell Biology, Department of Clinical and Experimental Medicine, Linköping University, SE-581 83 Linköping, Sweden

**Keywords:** bone metastasis, bone remodeling, epithelial-to-mesenchymal transition, pressure, adipose tissue-derived mesenchymal stromal cell

## Abstract

Once prostate cancer cells metastasize to bone, they perceive approximately 2 kPa compression. We hypothesize that 2 kPa compression stimulates the epithelial-to-mesenchymal transition (EMT) of prostate cancer cells and alters their production of paracrine signals to affect osteoclast and osteoblast behavior. Human DU145 prostate cancer cells were subjected to 2 kPa compression for 2 days. Compression decreased expression of 2 epithelial genes, 5 out of 13 mesenchymal genes, and increased 2 mesenchymal genes by DU145 cells, as quantified by qPCR. Conditioned medium (CM) of DU145 cells was added to human monocytes that were stimulated to differentiate into osteoclasts for 21 days. CM from compressed DU145 cells decreased osteoclast resorptive activity by 38% but did not affect osteoclast size and number compared to CM from non-compressed cells. CM was also added to human adipose stromal cells, grown in osteogenic medium. CM of compressed DU145 cells increased bone nodule production (Alizarin Red) by osteoblasts from four out of six donors. Compression did not affect IL6 or TNF-α production by PC DU145 cells. Our data suggest that compression affects EMT-related gene expression in DU145 cells, and alters their production of paracrine signals to decrease osteoclast resorptive activity while increasing mineralization by osteoblasts is donor dependent. This observation gives further insight in the altered behavior of PC cells upon mechanical stimuli, which could provide novel leads for therapies, preventing bone metastases.

## 1. Introduction

Prostate cancer (PC) is the second most diagnosed cancer worldwide, accounting for 375.000 deaths in 2020 [1]. The 5-year survival rate of primary PC tumors is nearly 100% due to effective treatments at this stage but reduces to 30% once PC cells metastasize to distant parts of the body [2]. The most common site of PC metastasis is bone, leading to either osteolytic lesions where excessive bone resorption results in a decline of bone mass, leading to pain, fractures and hypercalcemia or osteosclerotic lesions, where dysfunctional bone is synthesized [3,4,5]. PC cells are able to rearrange the tumor-bone microenvironment by secreting a plethora of paracrine factors, such as interleukin 6 (IL6), that modulate the activity of bone-forming osteoblasts, and bone-resorbing osteoclasts [6,7,8,9,10,11,12,13]. Besides biochemical signals, the tumor-bone microenvironment also provides metastasized PC cells with mechanical stimuli, such as strain, stress, and compressive forces resulting from daily activities and/or tumor progression [14]. Mechanical stimuli, which are omnipresent in bone, strongly influence the activity and communication of most, if not all, cells in bone and bone marrow, thereby governing the formation, adaptation, and regeneration of bone [15,16,17]. It is highly likely that these mechanical stimuli also modulate the communication between PC cells, osteoblasts, and osteoclasts to further rearrange the bone-tumor microenvironment and induce bone lesions, but to what extent is hereto unknown.

During progression of bone metastases, uncontrolled cell division and secreted extracellular matrix components continuously enhance the size and rigidity of the tumor [14]. The bone microenvironment in which these metastases reside is stiff and inelastic, which opposes tumor growth, resulting in compressive forces exerted on the tumor cells [18,19]. DU145 PC cells that were injected in the tibia of mice resulted in bone metastases that experienced an initial compression of 5 kPa, which stabilized at approximately 2 kPa after 20 days [20]. Compression of cancer cell types, other than PC cells, has already shown to profoundly affect tumor cell biology in vitro. Compressing cultured breast cancer cells, which also favor the bone as site for metastases, with a weighed piston corresponding to 0.7 kPa, induces cytoskeletal remodeling and alters their gene expression in favor of a more mesenchymal phenotype, a phenomenon referred to as epithelial-to-mesenchymal transition (EMT), resulting in cells with a higher metastatic and invasive potential [21,22]. Metastasized PC cells in bone may, similar to breast cancer cells, also become more mesenchymal and aggressive due to compression from the bone-tumor microenvironment. 

Compressing cancer cells also affects their production of paracrine signals (e.g., proteins and exosomes), thus presumably their communication with other cells in close proximity. For example, compressing breast cancer cells with 0.7 kPa enhances their vascular endothelial growth factor (VEGF) expression [23]. Compressing neuroglioma cells with 0.5 kPa induces the expression of growth and differentiation factor 15 (GDF15), a cytokine known to affect both osteoblast and osteoclast behavior [24,25]. For PC cells, little is known about the effect of compression on autocrine and paracrine signaling. It may be that the production of paracrine signals by PC cells are also modulated by compression, similar as for other cancer cell types, to interfere with normal bone remodeling and thereby contributing to the formation of bone lesions. We therefore hypothesize that compressing PC cells with 2 kPa stimulates EMT-related gene expression and alters their production of paracrine signals to affect the fate and functioning of osteoclasts and osteoblasts. 

## 2. Results

### 2.1. Compression Decreases Total DNA in DU145 Cell Cultures

To gain insight in whether compression affects PC cell number, e.g., due to cell death or impaired proliferation, we quantified Total DNA as a measure for cell number and *KI67* gene expression as an indication for proliferation. Compressing DU145 cell cultures with 2 kPa for 2 days tends to decrease Total DNA (2.73-fold, Figure 1A, *p* = 0.06) but did not affect *KI67* gene expression (Figure 1B) compared to non-compressed DU145 cell cultures. To gain insight in whether compression affects protein production per PC cell, we quantified Protein/DNA in DU145 cell cultures. Compressing DU145 cell cultures with 2 kPa for 2 days enhanced Protein/DNA 2.77-fold compared to non-compressed DU145 cell cultures (Figure 1C).

### 2.2. Compression Affects EMT-Related Gene Expression in DU145 Cells

Next, we aimed to gain insight into the effect of compression on the expression of EMT-related genes. We chose a multitude of epithelial and mesenchymal genes that are regulated during EMT, according to literature (full rationale for these genes can be found in supplementary paragraph 1). We further divided these genes into intracellular EMT-related genes (such as transcription factors and receptors; Figure 2A), and EMT-related signaling molecules (such as cytokines; Figure 2B) that can be secreted by PC cells to contribute to EMT in an autocrine or paracrine manner.

Compression decreased expression of both intrinsic epithelial phenotype-related genes by DU145 cells compared to non-compressed DU145 cells (*CDH1*: 3.41-fold and *SYND1*: 4.51-fold; Figure 2A). Compression also decreased expression of 2 out of 5 intrinsic mesenchymal phenotype-related genes by compressed DU145 cells compared to non-compressed DU145 cells (*Twist*: 9.59-fold and *Slug*: 1.79-fold; Figure 2A), but increased *Snail* gene expression 1.69-fold (Figure 2A). Compression did not significantly affect intrinsic mesenchymal phenotype-related expression of *VMN* or *PPARγ* in DU145 cells compared to non-compressed DU145 cells (Figure 2A).

Compression strongly decreased expression of EMT-related signaling molecules *TGF-β1*, *IL6* and *PTHrP* by compressed DU145 cells compared to non-compressed DU145 cells (3.16-fold, 22.17-fold and 10.25-fold, respectively; Figure 2B), but increased *GDF15* gene expression 1.49-fold (Figure 2B). Compression did not significantly affect the expression of EMT-related signaling molecules *TGF-β2*, *TNF-α*, *CYR61* or *WNT5a* by DU145 cells compared to non-compressed DU145 cells (Figure 2B).

### 2.3. Conditioned Medium from Compressed DU145 Cells Decreases the Resorptive Activity of Osteoclasts

Subsequently, we investigated the effect of conditioned medium (CM) from compressed and non-compressed DU145 cells on the formation and activity of osteoclasts. CM from non-compressed and from compressed DU145 cells both decreased the number of osteoclasts compared to fresh, unconditioned control medium (2.02-fold and 2.95-fold, respectively; Figure 3A,C). CM from non-compressed and compressed DU145 cells also both increased the cell area of osteoclasts compared to the control medium (3.98-fold and 2.56-fold, respectively; Figure 3A,D). CM from non-compressed DU145 cells did not affect the resorptive activity of osteoclasts compared to the control medium (Figure 3B,E), but the CM from compressed DU145 cells decreased the resorptive activity of osteoclasts by 38% compared to the control medium (Figure 3B,E).

### 2.4. Conditioned Medium from Compressed DU145 Cells Stimulates Bone Nodule Production by Osteogenic-Differentiated hASCs from the Majority of Donors

To investigate whether conditioned medium (CM) from (non-)compressed DU145 cells affect the differentiation of human mesenchymal stromal cells towards osteoblasts and subsequent production of bone-like matrix, CM was added to hASCs that were stimulated to differentiate into osteoblasts, followed by quantification of bone nodules (hASCs that were stimulated to differentiate into osteoblasts are hereon referred to as O-hASCs). CM from non-compressed DU145 cells slightly increased bone nodule production in the O-hASC culture from donor 6 compared to the control medium (+0.075 mM Alizarin Red concentration), but not from donor 1, 2, 3, 4 or 5 (Figure 4B). CM from compressed DU145 cells increased bone nodule production in the O-hASC culture from donor 1, 3, 4 and 6 compared to the control medium (+0.13 mM, +0.17 mM, +0.36 mM and +1.14 mM Alizarin Red concentration, respectively), but not from donor 2 and 5 (Figure 4B). CM from compressed DU145 cells increased bone nodule production in the O-hASC culture from donor 1, 3, 4 and 6 compared to CM from non-compressed DU145 cells (+0.78 mM, +0.20 mM, +0.40 mM and +1.07 mM Alizarin Red concentration, respectively), but not from donor 2, and 5 (Figure 4B).

### 2.5. Conditioned Medium from DU145 Cells Contains More IL6 Compared to the Control Medium 

We quantified protein concentrations of TNF-α and IL6 in the CM from (non)-compressed DU145 cells, because these proteins can affect EMT in PC cells, and affect osteoclast as well as osteoblast behavior. The protein concentration of TNF-α in all media was similarly low (CM from non-compressed DU145 cells: 1.28 pg/mL and CM from compressed DU145 cells: 1.03 pg/mL; no significant difference, Figure 5A). The protein concentration of IL6 in the CM from non-compressed DU145 cells (47.54 pg/mL) and in the CM from comrpessed DU145 cells (35.29 pg/mL) was much higher than in the unconditioned control medium (2.18 pg/mL), Figure 5B. The protein concentration of IL6 was similar in the CM from non-compressed and compressed DU145 cells, Figure 5B.

## 3. Discussion

We assessed whether 2 kPa compression, similar to the magnitude of compression on bone tumors generated by DU145 PC cells in mouse bone in vivo, affected the behavior of DU145 cells, a commonly used PC cell line to study metastasized prostate cancer. Compression tended to decrease Total DNA (*p* = 0.06) but enhanced Protein/DNA in DU145 cell cultures, and affected gene expression of EMT-related molecules. The conditioned medium (CM) of compressed or non-compressed DU145 cells induced fewer but larger osteoclasts and contained a higher concentration of IL6 protein compared to control medium. However, only the CM of compressed DU145 cells decreased the resorptive activity of osteoclasts, while increasing bone nodule formation by osteogenic-differentiated hASCs from 4 out of 6 human donors. This suggests that the mechanical environment has a major impact on the survival and behavior of metastasized PC cells, specifically their communication with effector cells in bone, i.e., osteoclasts and osteoblasts.

Major clinical challenges emerge once PC cells invade bone as there are no effective treatments at this stage of metastases, leading to drastically reduced survival rates, accompanied by severe osteolytic or osteosclerotic bone remodeling [26]. It is not fully understood which process in bone-tumor biology orchestrates bone remodeling. We therefore aimed to delineate the role of compression, induced by tumor expansion within bone, on PC cell behavior and their communication with bone cells. We show that mechanical compression likely induces a selection of PC cells to die as we found that compression decreased Total DNA in PC cells from all samples compared to non-compressed cells, while it did not affect gene expression of the proliferation marker *KI67*. KI67 is expressed when cells enter the cell cycle [27], indicating that the lower amount of DNA in compressed PC cell cultures is a result of apoptosis or necrosis rather than impaired proliferation. 

The PC cells that did survive compression in our in vitro-model, showed changes in expression of EMT-related genes. Specifically, two genes related to a more epithelial phenotype were decreased by PC cells in response to compression (*CDH1* and *SYND1*), while expression of the mesenchymal-related gene Snail and GDF15 was increased, which would indicate a more mesenchymal phenotype. However, other genes that have been linked to a more mesenchymal-phenotype (*Twist, Slug*, *TGF-β1*, *IL6* and *PTHrP*) were decreased by compressed PC cells. Thus, whether or not the EMT-phenotype really has occurred cannot be concluded from the data in this study, especially since we only quantified EMT-related gene expression and not protein (except for IL6 and TNF-α). 

We also showed that the paracrine signals in the CM from both compressed and non-compressed PC cells induces fewer, but larger osteoclasts compared to unconditioned medium. However, only the CM from compressed PC cells decreased the resorptive activity of osteoclasts, while it increased bone nodule production by 4 out of 6 O-hASC donors, compared to CM from non-compressed PC cells. This implies that mechanical compression, resulting from tumor expansion in bone, stimulates PC cells to produce paracrine signals that decrease bone resorption by osteoclasts but increase bone formation by osteoblasts from some but not all patients, leading to predominantly osteosclerotic tumors. To gain insight in which paracrine signals could be related to this effect, we quantified protein concentrations of TNF-α and IL6, which are two known modulators of osteoclast and osteoblast fate and functioning [28,29], in the CM from compressed and non-compressed PC cells as well as in the unconditioned control medium. IL6 can stimulate osteoclast formation when produced in large quantities (e.g., during inflammation), but also inhibit osteoclast formation [30]. We found higher IL6 protein concentrations in the CM from compressed and non-compressed PC cells compared to the control medium, which could therefore be related to decreased osteoclast formation while increasing in size. However, the decreased resorptive activity of osteoclasts in the CM from compressed PC cells compared to the control medium cannot not be explained by IL6, as IL6 protein concentrations where similar in the CM from non-compressed and compressed DU145 cells, while osteoclasts in CM from non-compressed PC cells did not show a decrease in resorptive activity. Therefore, other paracrine signals in the CM from compressed PC cells must account for the decrease in reportion by osteoclasts. Osteoblast formation has also shown to be inhibited by IL6 [13]. We found much higher IL6 protein concentrations in the CM from non-compressed and compressed PC cells compared to the control medium, while O-hASCs in the CM from compressed PC cells did not show a decrease in bone nodule production. Therefore, the increased bone nodule production by PC cells cannot be related to IL6. We did not find any significant difference in protein expression of TNF-α between different media. 

The changes in osteoclast and O-hASC behavior when cultured in the CM from non-compressed or compressed PC cells may also be related to other secreted factors that we quantified in this study (qPCR) which can affect osteoclast and osteoblast behavior. Osteoclast formation has been found to be inhibited by PTHrP [31]. We found lower *PTHrP* gene expression in compressed PC cells compared to non-compressed PC cells which therefore likely not contributes to decreased osteoclast formation and resorptive activity. Osteoclast formation has been found to be increased by TGF-β1 [32]. The lower expression of *TGF-β1* that we found in compressed PC cells may therefore contribute a decrease in osteoclast formation and resorptive activity. However, one does need to consider that TGF-β1 (and TGF-β2 for that matter) first need to bind with Small and Large Latent Complexes before being excreted by cells in order to exert any biological function [33]. It is possible that compression affects release of the active form. One study found that osteoclast formation was decreased by GDF15 [34] while another study found increased osteoclast formation when subjected to GDF15 and CM containing GDF15 from LNCaP cells [35]. LNCaP cells are another widely used cell line to assess prostate cancer. The CM from the compressed PC cell line; DU145 used in this study may therefore, similar to LNCaP cells, also decrease osteoclast formation via increased GDF15 expression. 

Osteoblast formation has been found to be increased by PTHrP overexpression in the prostate cancer cell line ACE-1 [36]. We found less *PTHrP* expression in compressed PC cells compared to non-compressed PC cells which can therefore not account for the increased mineralization by O-hASCs. GDF15 has been found to increase bone nodule formation by bone marrow cells [37]. Another study found that bone marrow cells increase the formation of bone nodules when subjected to GDF15 [37]. We found higher *GDF15* expression in compressed compared to non-compressed PC cells which may therefore be related to increased bone nodule formation by O-hASCs.

Additionally, TGF-β1 has shown to increase osteoblast formation and matrix deposition [38]. Therefore, the lower expression of *TGF-β1* that we found in compressed PC cells compared to non-compressed PC cells, cannot account for increased bone nodule formation by O-hASCs. 

Although we only quantified gene expression for most of the EMT-related signaling molecules by PC cells with qPCR, our results suggest that it is possible that altered expression of TGF-β1, PTHrP and GDF15 by compressed PC cells compared to non-compressed PC cells might be related to the changes in osteoclast and osteoblast behavior. 

All EMT-related signaling molecules that are assessed in this study are also expressed by cells within or surrounding the bone metastatic niche. TGF-β and PTHrP promote EMT and are secreted by osteoblasts [39,40,41,42]. IL6 promotes EMT and is secreted by osteoblasts and B-cells [43,44,45]. TNF-α promotes EMT and is secreted by macrophages, natural killer cells and T lymphocytes [46,47]. GDF15 promotes EMT and is secreted by osteocytes [37,48]. WNTs are involved in EMT and are produced by mesenchymal bone marrow cells [49]. Next to the possible paracrine, and EMT-affecting, signaling from cells within or around the bone metastatic niche to PC cells, compressed PC cells themselves also show altered gene expression of *TGF-β1*, *PTHrP, IL6* and *GDF15* as well as altered IL6 protein secretion in this study, which have the potential to affect their EMT-phenotype in an autocrine fashion, as well as altering the behavior of cells in the metastatic niche such as osteocytes and marrow stromal cells via paracrine signaling. Therefore, both biochemical and mechanical stimuli from the bone metastatic niche likely affect EMT in PC cells and their communication with other cells within or surrounding the bone metastatic niche and vice versa.

### Limitations

Translation of our observations to the in vivo situation has to be done with care: DU145 cells are often described as osteolytic, as injection of these cells leads to bone resorption in mice [50]. The effect of compression on other, more osteosclerotic PC cell types such as C4-B2, may differ and induce different PC behavior. In addition, DU145 cells, and other PC cell types, secrete enzymes that degrade bone such as cathepsin B and MMPs which can induce osteolytic bone lesions without affecting osteoclast or osteoblast biology [51,52,53]. Finally, we have only assessed the effect of mechanical compression while in vivo multiple mechanical (e.g., compression, shear stress, and environmental stiffness) and biochemical stimuli from the bone-tumor microenvironment affect tumor biology. Taken together, we have to be careful with extrapolating our results to the in vivo situation, but it is clear that compression, as occurs on bone metastases, strongly affects PC cell behavior. 

Our study has a number of other limitations as well. First, our model that simulates compression on PC cells is 2-dimensional (2D), resulting in a 2 kPa compression only on top of the cells. In vivo, compression within the tumor microenvironment is perceived in 3 dimensions (3D) which may have a different effect on cell behavior. For example, DU145 cells that were cultured in 2D showed increased proliferation compared to 3D [54]. Secondly, we have added our CM from PC cells to O-hASCs and osteoclasts that were cultured in the absence of mechanical stimuli. It may be that the effect of our CM from PC cells on osteoblasts and osteoclasts is different in the presence of mechanical stimuli that closely mimic the mechanical environment in bone. It should be noted, however, that everyday movement is unlikely in patients with advanced metastasis, where brittleness and pain often lead to bed rest and thus bone unloading and ablation of mechanical stimuli. We cannot be completely certain whether the produced bone nodules by osteoblasts in CM of compressed PC cells are a result of increased osteoblast activity or increased osteogenic differentiation of hASCs. It might be that the CM from non-compressed PC cells inhibits the osteogenic differentiation of hASCs (although the same amount of osteogenic-differentiation stimulating factors were added to all groups), while enhancing osteoblast activity if added to mature osteoblasts. Fourth, DU145 cells were not extracted from bone but from the brain [55]. One could therefore argue that the chemical and mechanical microenvironment of bone metastases, such as substrate stiffness, affected the phenotype of PC cells prior to isolation. Another limitation of this cell line is that they are androgen receptor (AR) negative while upregulation and activation of this receptor is often found during advanced stages of PC metastasis [56,57]. AR is likely involved in tumor progression and related to poor prognosis [56,57]. AR can be activated by any androgenetic hormone such as testosterone [58] but also by IL6 [59], which we found to be expressed by PC cells in this study. Therefore, the DU145 cell line in this study may not represent all forms of PC metastasis. It would be interesting to assess the effect of compression on AR positive PC cells although DU145 cells are considered a standard PC cell lines to assess metastases and have been shown to be a useful pre-clinical model [60]. 

## 4. Materials and Methods

### 4.1. Cell Cultures

DU145 cells were purchased from the American Type Culture Collection (ATCC, Manasas, VA, USA; Cat# HTB-81, RRID: CVCL_0105) and cultured in Eagle’s Minimum Essential Medium (EMEM; ATCC), supplemented with 10% Fetal Bovine Serum (FBS; Biowest, Nuaillé, France) and 1% antibiotics (PSF), containing 100 U/mL penicillin (Sigma-Aldrich, Bleiswijk, the Netherlands), 100 µg/mL streptomycin (Sigma-Aldrich) and 250 ng/mL amphotericin B (Sigma-Aldrich). Human Adipose Stem cells (hASCs) were collected from subcutaneous adipose tissue (surgical waste material) of six healthy donors (age: 31–36 years) during plastic surgery at the Tergooi Hospital Hilversum, The Netherlands as previously described [61]. Informed consent was obtained from all patients and The Ethical Review Board of the VU University Medical Center, Amsterdam, The Netherlands, approved the protocol (Protocol number: 2016/105) on 17 March 2016. Isolated hASCs were cultured in Minimum Essential Medium α (αMEM; Thermo Fisher Scientific, Waltham, MA, USA), supplemented with 10% Fetal Clone Serum 1 (FC1; HyClone, Logan, UT, USA) and 1% PSF (Sigma-Aldrich). 

### 4.2. Compression on Prostate Cancer Cells

We applied compression on DU145 cells by modifying a weight-approach based setup that was originally described by Kalli et al. [62]. Briefly, DU145 cells were seeded (1 × 10^5^ cells/cm^2^) in 6 well inserts (Greiner, Alphen aan den Rijn, the Netherlands; pore size: 3 µm, pore density: 0.6 × 106/cm^2^) within EMEM (ATCC), 10% FBS (Biowest), and 1% PSF (Sigma-Aldrich) and incubated for 24 h at 37 °C and 5% CO_2_. Subsequently, agarose cushions were placed on top of the cells. The agarose cushions were made by first mixing 6 g UltraPure Low Melting Point Agrose (Thermo Fisher Scientific) with 100 mL distilled H_2_O resulting in agarose gels. These agarose gels were then melted, cooled to 40 °C and mixed (ratio 1:1) with pre-heated medium (37 °C) containing a double concentration of EMEM (Corning, New York, NY, USA), 20% FBS (Biowest), and 2% PSF (Sigma-Aldrich), followed by 5 min cooling off to ensure solidification of the formed agarose cushions. On top of the agarose cushions, customized 100 g cylindrical stainless steel (type: 304) weights with Teflon encasing were placed (van Gelder, Utrecht, the Netherlands). These weights exerted a 2 kPa compression on the cells, which were then incubated for 48 h at 37 °C and 5% CO_2_. Thereafter, compressed and non-compressed DU145 cells were lysed in 700 µL TRIzol (Thermo Fisher Scientific) reagent per sample and stored in −80 °C while conditioned medium (CM) was11isuo collected11isualized11slyy and stored at −20 °C. 

### 4.3. RNA Isolation

Total RNA was isolated from the 700 µL TRIzol (per sample) containing the cell lysates from compressed or non-compressed DU145 cells (*n* = 20) according to the instructions provided by Thermo-Fisher Scientific, Waltham, MA, USA [63]. RNA was dissolved in 30 µL DEPEC-treated H_2_O (Thermo-Fisher Scientific) and quantified by reading the absorbance of 2 µL dissolved RNA at 260 nm using a Synergy HT spectrophotometer (BioTek Instruments, Winooski, VT, USA). RNA purity was assessed by dividing the absorbance of each sample at 260 nm over 280 nm (A_260_/A_280_ ratio), using the same Synergy HT spectrophotometer (BioTek Instruments) and considered pure when A_260_/A_280_ > 1.8. 

### 4.4. cDNA Synthesis 

A total of 750 ng RNA per sample was complemented with DEPEC-treated H_2_O (Thermo-Fisher Scientific) until a final volume of 9 µL and reversed transcribed to cDNA using a RevertAid First Strand cDNA Synthesis Kit (Fermentas, St. Leon-Rot, Germany) according to the manufacturer’s instructions. Briefly, 1 µL Oligo (dT) Primer and 1 µL Random Hexamer Primer (50 µM) were added to the 9 µL of 750 ng RNA and incubated at at 65 °C for 5 min, followed by cooling on ice. Subsequently, 4 µL Reaction Buffer (5X), 1 µL RiboLock rNase inhibitor (20 U/µL), 2 µL dNTP mix (10 mM) and 1 µL RevertAid M-MuLV RT (200 U/µL) were added to construct a total volume of 20 µL which was then incubated at 42 °C for 60 min, 5 min at 25 °C, 42 °C for 60 min and finally at 70 °C for 5 min to terminate the reactions. cDNA samples were thereafter stored at −80 °C.

### 4.5. qPCR

Quantitative polymerase chain reactions (qPCR) were prepared with 5 μL SYBR Green I Mastermix (Roche Diagnostics, Basel, Switzerland), 4 μL cDNA, 0.5 μL forward primer (1 μM), and 0.5 μL reverse primer (1 μM) to a final volume of 10 µL within a 384 well plate (4 titude, Leiden, The Netherlands). qPCR was performed using a LightCycler 480 (Roche) using cycling conditions that included a polymerase activation step of 95 °C for 10 min and 45 cycles of 95 °C for 10 s and annealing/extension at 56 °C for 5 s with melt curve analysis from 56 to 96 °C in 0.6 °C increments. *PBGD*, *18S* and *B2M* were used as housekeeping genes as the cube root of their combined product was stable between conditions, calculated according to the methodology described by BestKeeper [64]**.** qPCR was conducted according to the Pfaffl method [65], using various concentrations of DU145 cDNA to generate a standard curve in order to quantify the concentration of *KI67*, *CDH1*, *SYND1*, *Slug*, *Twist*, *Snail*, *PPARγ*, *VMN*, *CYR61*, *TGF-β1*, *TGF-β2*, *PTHRP*, *IL6*, *TNF-α*, *GDF15* and *WNT5a*. The sequences of each primer are listed in Table 1 below. For each gene quantified via qPCR, undetectable samples were excluded from analysis. Outliers were also identified using Grubb’s test (aggressiveness = Q1) from GraphPad prism version 9 (GraphPad Software, San Diego, CA, USA) and also excluded from analysis. The number of undetectable samples and outliers for each gene are shown in Appendix A.

### 4.6. LEGENDplex Protein Quantification

Protein concentrations of TNF-α and IL6 in the CM from (non-)compressed DU145 cells and the unconditioned control medium (EMEM; ATCC, 10% FBS; Biowest, and 1% PSF; Sigma-Aldrich) were determined with a custom designed LEGENDplex Multi-Analyte Flow Assay Kit, bought from Biolegend (BioLegend, San Diego, CA, USA). The kit contains beads that attach specifically to TNF-α or IL6 as well as fluorescent antibodies that attach to the bead. TNF-α and IL6 labeled beads, together with varying bead concentrations (to generate the standard curve), were then quantified based on size and fluorescence intensity in duplo using a BD Accuri C6 Flow Cytometer (BD Biosciences, East Ruthford, NJ, USA). Analysis of the results (duplo mean for each sample) was performed with LEGENDplex software version 8 (BioLegend). Protein concentrations were calculated using the generated standard curve and expressed in picograms per mililiter (pg/mL). 

### 4.7. Total DNA and Protein/DNA Quantification

To quantify Total DNA and Protein/DNA in (non-)compressed DU145 cell cultures, we extracted DNA and Protein from 5 independent experiments (compressed and non-compressed, *n* = 5) according to the instructions provided by Thermo-Fisher Scientific [63]. To quantify Total DNA in DU145 cell cultures, we performed the CyQUANT Cell Proliferation Assay (Thermo-Fischer Scientific C7026) according the manufacturer’s instructions. Briefly, the DNA pellet of each DU145 cell culture was dissolved in 20 µL 8 mM NaOH and 80 µL deionized H_2_O. Then, 50 µL of each sample was incubated with 50 µL Nucleic Acid Stain on a black 96 well micro plate (Thermo-Fisher Scientific), together with various concentrations of DNA to generate the standard curve. Subsequently, fluorescence measurements of the stained DNA were read at 485 nm using a Synergy HT spectrophotometer (BioTek Instruments). Thereafter, Total DNA per DU145 cell culture was calculated using the standard curve and expressed in nanogram per milliliter (ng/mL). We quantified Total DNA in 50 µL per sample, and therefore divided Total DNA, as calculated by the standard curve in nanogram per milliliter, with 20 (1000 µL/20 = 50 µL sample) to obtain the Total DNA values per sample. Total DNA is depicted as Total DNA (ng) per sample.

To quantify Protein/DNA in DU145 cells cultures, we performed the Pierce BCA Protein Assay Kit (Thermo-Fisher Scientific 23225) according to the manufacturer’s instructions. Briefly, the Protein pellet of each DU145 cell culture was dissolved in 50 µL deionized H_2_O and 50 µL 1% Sodium Dodecyl Sulfate. After centrifugation at 10.000× *g* for 10 min, 40 µL of the supernatant from each sample was incubated on a 96 well plate (Thermo-Fisher Scientific) with the Kit’s reagents, together with various concentrations of Bovine Serum Albumin (BSA) to generate the standard curve at 37 °C and 5% CO_2_ for 15 min. Subsequently, the purple-colored reaction by chelation of 2 molecules BSA with one cuprous ion was read at 562 nm using a Synergy HT spectrophotometer (BioTek Instruments). Thereafter, Total Protein per DU145 cell culture was calculated using the standard curve and expressed in µg/mL. We quantified Total Protein in 40 µL per sample, and therefore divided Total Protein, as calculated using the standard curve microgram per milliliter with 25 (1000 µL/25 = 40 µL sample) to obtain Total Protein values per sample. Thereafter, Total protein per sample was normalized for Total DNA per sample and depicted as microgram Protein per nanogram DNA (µg/ng) per sample.

### 4.8. CD14^+^ Monocyte Isolation and Osteoclastogenesis

For osteoclast number and size quantification, one buffy coat from one patient was bought from Sanquin (Sanquin, Amsterdam, The Netherlands). For osteoclast resorption, performed on a separate occasion, 38 mL of blood was collected from two healthy volunteers, after obtaining informed consent. Peripheral blood mononuclear cells were isolated from the whole blood using Lymphoprep density gradient centrifugation (Alere Technologies AS, Oslo, Norway). Mononuclear cells obtained from blood and buffy coat were incubated with CD14-antibody tagged microbeads (Miltenyi, Biotec, Bergisch Gladbach, Germany) and sorted by a MACS magnetic cell sorter (Biotec) according to the instructions of the manufacturer [66]. Isolated CD14^+^ monocytes were counted with a Muse Cell analyzer (Merck, Burlington, MA, USA) according to the instructions of the manufacturer [67].

CD14^+^ monocytes were seeded in a 96 well plate (34 × 10^3^ cells/well) in Eagle’s Minimum Essential Medium with Alpha Modification (αMEM; Thermo Fisher Scientific), 10% Fetal Clone Serum 1 (HyClone), 1% PSF (Sigma-Aldrich) and 25 ng/mL macrophage colony-stimulating factor (M-CSF; R&D systems, Oxon, UK). After 3 days, the medium was replaced by the media below in Table 2. All media were mixed (ratio 1:1) with the osteoclast medium (αMEM; Thermo Fisher Scientific, supplemented with 10% FC1; HyClone, and 1% PSF (Merck). All media had a final concentration of 1% PSF (Sigma-Aldrich), 10 ng/mL M-CSF (R&D systems), and 2 ng/mL receptor activator of nuclear factor kappa-B ligand (RANKL; R&D systems). All media were replaced twice a week and cells within these media were incubated at 37 °C and 5% CO_2_ for 21 days.

### 4.9. TRACP and DAPI Staining

First, all cells were fixated with 4% formaldehyde (Sigma-Aldrich). Then, using a Leukocyte Acid Phosphatase staining kit (Sigma-Aldrich), Tartrate Resistant Acid Phosphatase (TRACP) staining was performed according to a method described previously [68]. Diamidino-2 phenylindole dihydrochloride (DAPI; Thermo-Fisher Scientific) was used for staining nuclei. An osteoclast was identified as a TRACP positive stained cell containing 3 or more nuclei, using bright field and fluorescent microscopy (Leica, Wetzlar, Germany). For each well, 6 microscopic images were taken (10× magnification) from the left side of the well to the right side of the well, spanning the middle horizontally. 

### 4.10. Resorption Assay

CD14^+^ monocytes from the two healthy volunteers were seeded (6.4 or 10 × 10^4^ cells/well) in 96 well Osteo Assay surface plates (Corning costar, Lowell, MA, USA) and cultured for 2 weeks in the media described in Table 2. Subsequently, cells were incubated with 10% bleach for 5 min, washed with H_2_O, and then air dried. For each well, 4 microscopic images (4× magnification) were taken with a Leica Ti Eclipse (Leica) to image the total well. Resorption was quantified as white area (resorbed Calcium-phosphate)/brown area (remaining Calcium-phosphate) × 100% and normalized to resorption of calcium phosphate in the control medium using ImageJ software (ImageJ, Bethesda, MD, USA). 

### 4.11. Osteoblastogenesis 

hASCs from 6 different donors were seeded in 12 well plates (40 × 10^3^ cells/well) containing Dulbecco’s Modified Eagle Medium (DMEM; Thermo Fisher Scientific), 2% human platelet lysate (PL; Merck), 10 IU/mL heparin (LEO Pharma A/S, Ballerup, Denmark) and 1% PSF (Sigma-Aldrich). After 24 h, the medium was replaced by the media described below in Table 3. All media were mixed (ratio 1:2) with fresh osteogenic medium (DMEM; Thermo Fisher Scientific, supplemented with 10 IU/mL; LEO Pharma, 2% PL; Merck, and 1% PSF; Merck). All media had a final concentration of 1% PSF (Sigma-Aldrich), 10 IU/mL heparin, 50 µM vitamin C (Sigma-Aldrich), 5 mM β-glycerophosphate (Sigma-Aldrich), and 10 nM 1α,25-Dihydroxy-vitamin D_3_ (Sigma-Aldrich). All media were replaced twice a week and cells within these media were incubated at 37 °C and 5% CO_2_ for 21 days. hASCs that were stimulated to differentiate into osteoblasts are referred to as osteogenic-differentiated hASCs (O-hASCs).

### 4.12. Mineralization Assay

The mineralized bone matrix deposited by O-hASCs was assessed using Alizarin Red staining (Merck) as described previously [69]. In short, O-hASCs were fixed in 4% formaldehyde for 15 min, then washed with H_2_O and incubated with Alizarin Red for 30 min at room temperature. Images were taken from the red bone nodules produced by O-hASCs with a HP Scanjet 4070 (Hewlett-Packard, Palo Alto, CA, USA). After the images were collected, cells and nodules were dissolved in 10% acetic acid (Merck) and centrifuged at 20,000× *g* for 15 min. Alizarin Red absorbance was read at 405 nm using a Synergy HT^®^ spectrophotometer (BioTek Instruments). Absorbance values of the samples were plotted against absorbance values of known Alizarin Red concentrations. 

### 4.13. Statistical Analysis

Two groups with normally distributed data and homogeneous variances were compared with paired *t*-tests when our samples were paired, e.g., loaded and unloaded samples derived from the same cells. When samples were missing, e.g., due to outlier identification, normally distributed data from two groups with homogeneous variances were compared with a standard *t*-test. Two groups with skewed data and/or heterogeneous variances were first log-transformed, whereafter *t*-tests were used for comparison as described directly above. If normality and homogeneity of variance were still violated after log-transformation, data were analyzed with Wilcoxon signed-rank tests or with Mann–Whitney U tests when paired testing was not applicable. Three groups with normally distributed data and homogeneous variances were compared with repeated measures ANOVA’s and paired *t*-tests as post hoc using Bonferroni corrections. Three groups with skewed data and/or heterogeneous variances (even after log-transformation) were compared using Kruskal–Wallis tests and Mann–Whitney U tests as post hoc using Bonferroni corrections. Differences were considered statistically significant at *p* < 0.05. All analyses were performed with rStudio version 4.0.3 (Rstudio, Boston, MA, USA) and all graphs were created with GraphPad Prism version 9 (GraphPad Software). Figure 1 and Figure 2 were visualized with individual data points and a horizontal bar which depicts the median. Figure 3 and Figure 5 were visualized with box-whisker plots plots which depict the lower extreme, lower quartile, median, upper quartile and upper extreme. Figure 4 was visualized with bar plots that depict the values of each individual donor separate. We consider one experimental unit (*n* = 1) when the experiment was conducted on a separate occasion from other experiments, or when cells from different donors were used. 

## 5. Conclusions

We conclude that compressing DU145 cells affects EMT-related gene expression, although whether a more epithelial or more mesenchymal phenotype was induced could not be concluded from our qPCR analysis. Compression of DU145 cells modulates their production of paracrine signals to decrease the resorptive activity of osteoclasts, while enhancing bone nodule formation by O-hASCs from 4 out of the 6 donors compared to non-compressed DU145 cells. These changes in osteoclast and O-hASC behavior are likely not related to IL6, indicating that other paracrine signals from compressed PC cells must account for the decreased resorptive activity of osteoclast and increase bone nodule formation by O-hASCs. Altered expression of *TGF-β1*, *PTHrP* and *GDF15* by compressed PC cells could be related to reducing the resorptive activity of osteoclasts, while enhancing bone nodule formation by O-hASCs, but future studies quantifying protein expression of these factors, and possibly blocking the production of these factors are requires to draw conclusions about causation. Elucidation of the molecular pathways downstream of sensing mechanical compression in PC cells might provide novel targets for pharmaceutical intervention.

## 6. Supplementary Paragraph 1: Rationale for Genes to Assess EMT

During epithelial-to-mesenchymal transition (EMT), cells lose their cell–cell adhesion complexes and polarity, while gaining invasive and migratory characteristics [70]. EMT is thought to be induced by signaling factors such as TGF-β, TNF-α and several WNTs, which activate pathways such as downstream TGF-β, AKT, and ERK pathways that initiate EMT-related gene expression in cancer cells [70,71]. In our study, we divided our EMT-related genes in two categories: (1) intrinsic EMT-related genes, coding for proteins in- and on the PC cells, and directly related to more mesenchymal cell characteristics such as decreased adhesion and increased invasion, and (2) EMT-related signaling factors, which can be produced by tumor cells, or other cells within the tumor niche, to activate EMT-related signaling pathways. 

### 6.1. Intrinsic EMT-Related Proteins

The most well-known hallmark for EMT is the downregulation of the epithelial cell–cell junction protein E-cadherin (CDH1) and upregulation of the transcription factors Snail, Slug and Twist that induce mesenchymal gene expression [70,72]. Another well-known epithelial gene is Syndecan-1 (SYND1) which is a proteoglycan that mediates cell adhesion to extracellular matrix components such as collagen [73]. Syndecan-1 expression is inversely correlated with tumor progression, invasion and migration as Syndecan-1 increases E-cadherin expression, but decreases Snail and Vimentin expression (VMN) [74].

Snail, Slug and Twist directly inhibit E-cadherin expression by binding to the E-box sequences in the E-cadherin promotor, leading to suppression of the tight and gap junctions between cells and subsequent increased invasion and migration potential [75,76,77,78]. These molecules are therefore associated with a more mesenchymal-like phenotype. Vimentin is an intermediate filament necessary for cell integrity and is involved in adhesion, and migration and a marker for mesenchymal cells [79]. Vimentin has been found to be highly expressed during metastases and associated with poor clinical outcomes [79,80]. Twist can upregulate CuL2 circular RNA which inhibit Vimentin-targeting miRNAs, leading to increased Vimentin expression [81]. Considering that Twist inhibits E-cadherin expression and increases Vimentin expression, Twist can be considered to be associated with a mesenchymal-like phenotype. PPARγ leads to PG1α localization to the nucleus which stimulates mitochondrial synthesis to fuel progression of EMT, leading to a more aggressive phenotype [82]. 

### 6.2. EMT-Related Signaling Molecules

Parathyroid hormone related protein (PTHrP) is expressed by PC cells as well as osteoblast and enhances EMT by upregulating Snail and Vimentin and downregulating E-cadherin [41,42]. Cystine-rich angiogenic inducer 61 (CYR61) is a matricellular protein involved in regulating adhesion and migration, while also functioning as a angiogenic factor [83]. CYR61 promotes EMT by upregulating Twist and decreasing E-cadherin via αvβ5 integrin, Raf-1, MEK, ERK, and ElK-1 signaling pathways [84]. Tumor necrosis factor alpha (TNF-α), secreted by cancerous cells, T lymphocytes and natural killer cells, induces Twist upregulation via the NF-κB signaling pathway which in then inhibits E-cadherin and increases Vimentin expression [46,47,85,86]. Interleukin 6 (IL6) expressed by cancerous cells, osteoblasts and B cells, can activate the AKT, ERK and STAT3 pathways, leading to increased expression of Snail, Twist and vimentin and a decreased expression of E-cadherin [87]. Growth and differentiation factor 15 (GDF15) is a member of the transforming growth factor family and together with TGF-β1/2, expressed by cancerous cells and osteoblasts, induce EMT via activation of the AKT, ERK, and Nf-κB pathways, leading to increased expression of Snail, Twist and Vimentin while inhibiting E-cadherin [37,86,87,88,89]. Wingless/Integrated gene 5a (WNT5a), expressed by cancerous cells and mesenchymal bone marrow cells, can induce EMT via activation of the AKT, ERK pathways that upregulate Twist and Vimentin and downregulate E-cadherin [49,90,91,92]. All of these soluble factors thus favor a more mesenchymal-like phenotype.

## Figures and Tables

**Figure 1 ijms-24-00759-f001:**
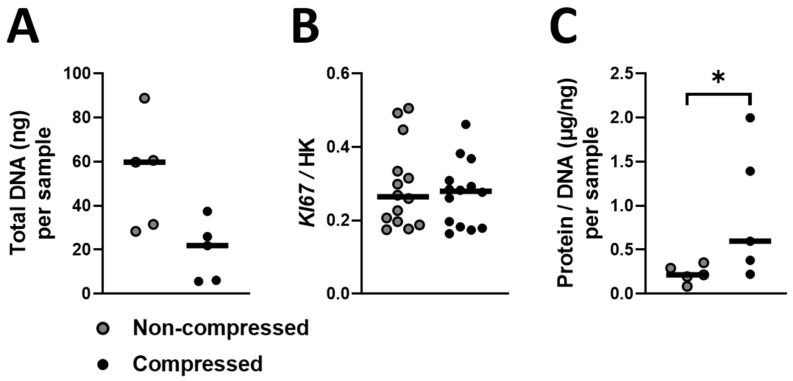
**Effect of 2 kPa compression on DU145 Total DNA, proliferation and Protein/DNA**. 5 separate cultures of DU145 cells were compressed with 2 kPa for 2 days. We then quantified (**A**) Total DNA using a CyQUANT Cell Proliferation Assay as a surrogate for cell number per sample (ng, *n* = 5), and (**B**) Total protein using a Pierce BCA Protein Assay Kit. Total protein per sample was divided through Total DNA per sample to obtain an impression of protein production per cell for each sample (µg/ng, *n* = 5). (**C**) In other experiments, a total of 20 separate cultures of DU145 cells were compressed with 2 kPa for 2 days and RNA was isolated. *KI67* gene expression was determined as an indication for cell proliferation using qPCR. *KI67* gene expression was detectable and quantifiable in 14 out of 20 samples (*n* = 14). (**A**,**B**) Analyzed with paired *t*-tests, (**C**) Analyzed with a paired *t*-test on log-transformed data. * *p* < 0.05.

**Figure 2 ijms-24-00759-f002:**
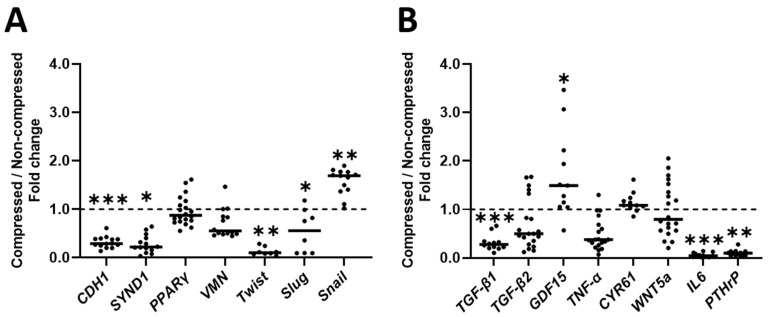
**Effect of compression on gene expression related to EMT in DU145 cells**. 20 separate DU145 cell cultures were compressed with 2 kPa for 2 days, followed by qPCR on (**A**) Gene expression of EMT-related factors expressed in- or on PC cells (*n* = 8–20), and (**B**) Gene expression of EMT-related signaling molecules (*n* = 11–20). *VMN*, *WNT5a* and *PPARγ* were analyzed with paired *t*-tests. *Slug*, *GDF15* and *TGF-β1* were analyzed with normal *t*-tests. *CDH1* was analyzed with a paired *t*-test on log-transformed data. *Snail* was analyzed with a normal *t*-test on log-transformed data. *CYR61*, *Twist*, *PTHrP*, *IL6* and *SYND1* were analyzed with Mann–Whitney U tests. *TGF-β2* was analyzed with a Wilcoxon signed-rank test.* *p* < 0.05, ** *p* < 0.01, *** *p* < 0.001.

**Figure 3 ijms-24-00759-f003:**
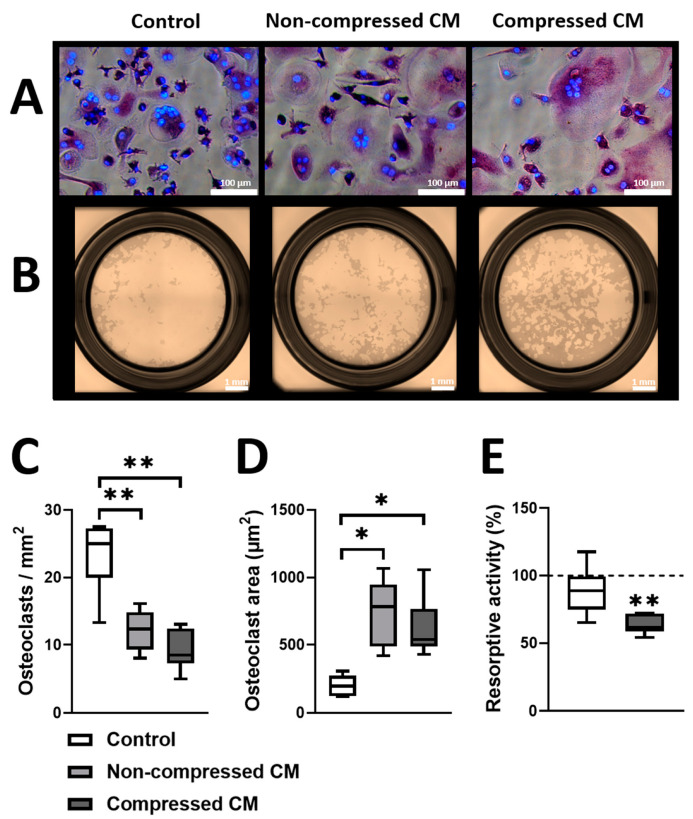
**Effect of CM from (non-)compressed DU145 cells on osteoclast number, size, and resorptive activity.** (**A**) Representative images (scale bar = 100 µm) of the number and area of osteoclasts when subjected to conditioned medium (CM) from DU145 cells that were either or not compressed with 2 kPa (mixed 1:1 with standard osteoclast medium). An osteoclast is defined as 3 or more nuclei (blue DAPI stained dots) within the periphery of a cell (purple TRACP staining). Control = osteoclasts, cultured in unconditioned PC medium (also mixed 1:1 with standard osteoclast medium). (**B**) Representative images (scale bar = 1 mm) of osteoclast resorptive activity. (**C**) Number of osteoclasts per mm^2^ (counted manually with ImageJ), (**D**) Osteoclast area (calculated using the freehand tool from ImageJ to outline individual cell peripheries), (**E**) Osteoclast resorptive activity (quantified as white area (resorbed Calcium-phosphate)/brown area (remaining Calcium-phosphate) × 100%). (**C**) Analyzed with a repeated measures ANOVA and paired *t*-tests post hoc, (**D**) Analyzed with a Kruskal–Wallis test and Mann–Whitney U tests post hoc, (**E**) Analyzed with paired *t*-tests on log-transformed data. * *p* < 0.05, ** *p* < 0.01 (*n* = 6).

**Figure 4 ijms-24-00759-f004:**
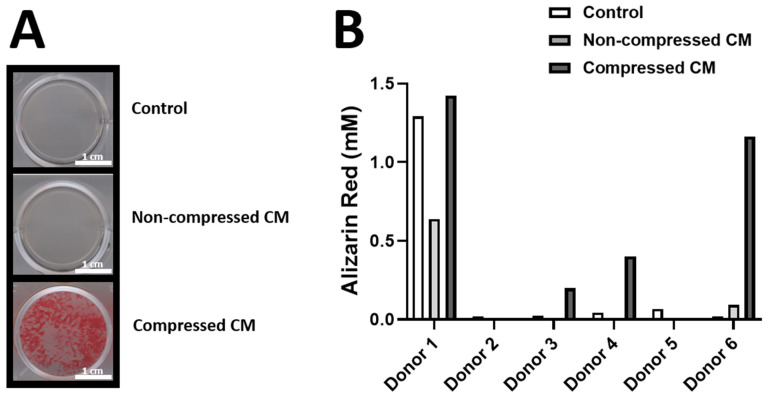
**Effect of CM from (non-)compressed DU145 cells on O-hASC mineralization.** (**A**) Representative Images of formed bone nodules (Alizarin Red staining) by O-hASCs (6 donors, *n* = 6) that were cultured in conditioned medium (CM) from DU145 cells that were either or not compressed with 2 kPa (mixed 1:2 with osteogenic medium) (scale bar = 1 cm). Control = O-hASCs, cultured in unconditioned PC medium (also mixed 1:2 with osteogenic medium). (**B**) Alizarin Red concentrations (mM) (calculated by plotting the Alizarin Red absorbance values at 590 nm using spectophotometry against the absorbance values of known Alizarin Red concentrations) corresponding to the formed bone nodules by O-hASCs from the 6 donors.

**Figure 5 ijms-24-00759-f005:**
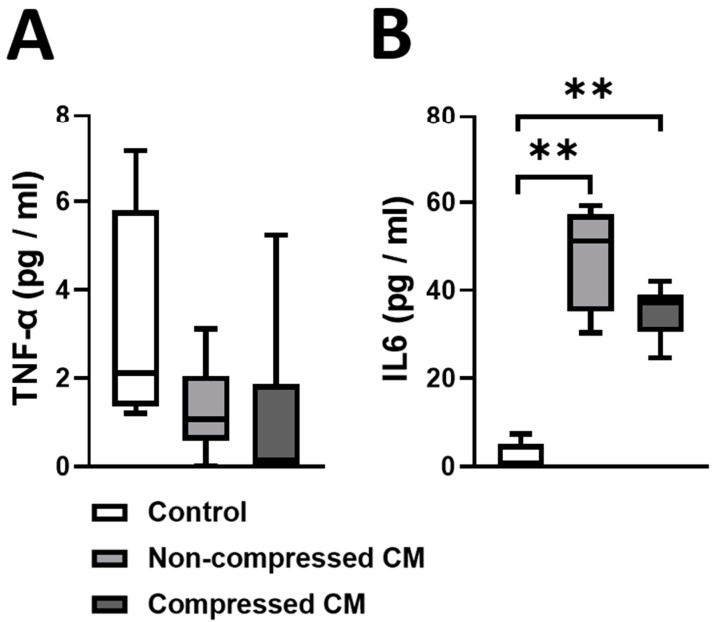
**Protein concentrations of TNF-α and IL6 in the CM from (non)-compressed DU145 cells.** DU145 cells were compressed with 2 kPa for 2 days. Subsequently, using a LEGENDplex Multi-Analyte Flow Assay Kit, we quantified (**A**) Protein concentrations of TNF-α (pg/mL) and (**B**) IL6 (pg/mL) in the CM from compressed and non-compressed DU145 cells as well as in the unconditioned control medium. (**A**,**B**) Analyzed with a Kruskal–Wallis test and Mann–Whitney U tests post hoc. ** *p* < 0.01 (*n* = 6).

**Table 1 ijms-24-00759-t001:** Primer sequences.

*Gene* (*Human*)	Forward	Reverse
** *PTHrP* **	GCCGTAAATCTTGGATGGACTT	AGCGTCGCGGTGTTCCT
** *SYND1* **	GCCACCATGAGACCTCAACC	CGGCCACTACAGCCGTATTC
** *Slug* **	ACTACAGCGAACTGGACAC	GATGAGGAGTATCCGGAAAG
** *Twist* **	GGCCAGGTACATCGACTTCC	TATCCAGCTCCAGAGTCTCTA
** *IL6* **	ACAGCCACTCACCTCTTCA	ACCAGGCAAGTCTCCTCAT
** *TNF-α* **	AGAGGGCCTGTACCTCATCT	AGGGCAATGATCCCAAAGTAG
** *WNT5a* **	GTGTCGCTAGGTATGAATAA	CGCGTATGTGAAGGCCGTC
** *PPARγ* **	CGACCAGCTGAATCCAGAGT	GATGCGGATGGCCACCTCTT
** *TGF-β1* **	CTACTACGCCAAGGAGGTCA	CACGTGCTGCTCCACTTT
** *TGF-β2* **	GGCACCTCCACATATACCAGT	ATTTCCACCCTAGATCCCTCTT
** *GDF15* **	CCTGCAGTCCGGATACTCAC	CGAAGATTCTGCCAGCAGTT
** *CYR61* **	TCCGAGGTGGAGTTGACGAGAA	TTCACAAGGCGGCACTCAGG
** *Snail* **	GGCTGCTACAAGGCCATGTC	CGCCTGGCACTGGTACTTCT
** *KI67* **	CCCTCAGCAAGCCTGAGAA	AGAGGCGTATTAGGAGGCAAG
* ** *VMN* ** *	TGCGTCTCTGGCACGTCTTGA	CAGGTTCTTGGCAGCCACACT
* ** *CDH1* ** *	CTCCTGGCCTCAGAAGACA	GTGGCAATGCGTTCTCTATC
** *18S* **	GTAACCCGTTGAACCCATT	GTAACCCGTTGAACCCATT
** *PBGD* **	TGCAGTTTGAAATCATTGCTATGTC	AACAGGCTTTTCTCTCCAATCTTAG
** *B2M* **	AGCTGTGCTCGCGCTACTCTC	CACACGGCAGGCATACTCATC

*PTHRP*, Parathyroid hormone like hormone; *SYND1*, Syndecan 1; *Slug*; *IL6*, Interleukin 6; *TNF-α*, Tumor Necrosis Factor alpha; *WNT5a*, Wingless/Integrated gene 5a; *PPARγ*, peroxisome proliferator activated receptor gamma; *TGF-β2*, Transforming Growth Factor Beta 1; *TGF-β2*, Transforming Growth Factor Beta 2; *GDF15*, Growth Differentiation Factor 15; *Twist*, Twist Family BHLH Transcription Factor 1; *CYR61*, Cysteine-rich angiogenic inducer 61; *Snail*, Zinc finger protein SNAI1; *KI67*, Nuclear protein 67; *VMN*, Vimentin; *CDH1*, E-Cadherin, *18s*, 18 Svegdber units; *PBGD*, Porphobilinogen deaminase; *B2M*, β2 microglobulin.

**Table 2 ijms-24-00759-t002:** Osteoclastogenesis media.

Medium	Conditioned by PC (+/− Compression)	Basis	Supplement
Osteoclast medium	No	αMEM	FC1 (10%)
PC medium unconditioned (control)	No	EMEM	FBS (10%)
CM non-compressed PC cells	Yes	EMEM	FBS (10%)
CM compressed PC cells	Yes	EMEM	FBS (10%)

PC, Prostate Cancer; CM, Conditioned Medium; αMEM, Eagle’s Minimum Essential Medium with Alpha Modification; EMEM; Eagle’s Minimum Essential Medium; FC1, Fetal Clone Serum 1; FBS. Fetal Bovine Serum.

**Table 3 ijms-24-00759-t003:** Osteoblastogenesis media.

Medium	Conditioned by PC (+/− Compression)	Basis	Supplement
Osteoblast medium	No	DMEM	PL (2%)
PC medium unconditioned (control)	No	EMEM	FBS (10%)
CM non-compressed PC cells	Yes	EMEM	FBS (10%)
CM compressed PC cells	Yes	EMEM	FBS (10%)

PC, Prostate Cancer; CM, Conditioned Medium; DMEM, Dulbecco’s Modified Eagle Medium; EMEM; Eagle’s Minimum Essential Medium; Pl, Platelet Lysate; FBS. Fetal Bovine Serum.

## Data Availability

All data in this manuscript is the intellectual property of the Academic Centre for Dentistry Amsterdam (ACTA). All data is available from the corresponding author Dr. Astrid D. Bakker, associate professor at ACTA.

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
