# Peer review of "Compressed Prostate Cancer Cells Decrease Osteoclast Activity While Enhancing Osteoblast Activity In Vitro"

_ijms, 2023, doi:10.3390/ijms24010759_

Round 1
Reviewer 1 Report (New Reviewer)
The authors presented an elegant paper about a prostate cancer cell line subjected to compression and how it can affect its functions. Through compression of prostate cancer cells, they showed a reduction in osteoclasts' resorption activity, which can explain the osteoblastic nature of most prostate metastasis. The English is good, and only minor mistakes can be corrected. The methods are well explained and designed, though the DU145 cell lineage is questionable because, as mentioned by the authors, it has an osteolytic nature. The results are clear, as well as the figures, and straightforward. The introduction and discussion are well-written and with important citations about the subject. However, I have one comment here before publication. I think both are too long and should be shortened, especially the introduction session. In general, the work is good and is almost ready to be published.
Author Response
Please see attachment

Reviewer 2 Report (New Reviewer)
The authors conducted a study with a very suitable methodology. However, some points caused me fear.
First, the statistics described for analysis of more than two groups mention only data with a Normal distribution (ANOVA), but there is a test for data with asymmetric distribution when only two groups are used. Did the data from more than one group all have a Normal distribution? Has equality of variance and all other ANOVA requirements been tested? I know that the sample n is small due to the nature of the study, but these questions must be carried out with the aim of critical reflection. I suggest implementing a post hoc Kruskal-Wallis test and comparing the p-values with the currently obtained ones.
Still on statistical analysis, even though it is common in the area of molecular biology and others with a small sample size, the representation of bar graphs with standard error of the mean is not adequate, since confidence intervals are not elaborated, but the standard deviation is. Also, box-plot or violin plots are better suited than bar plots for visual and comparative analysis of data distributions.
As for the analysis of bone nodule production, a visual and empirical analysis can lead to many errors due to the lack of statistics. I suggest normalizing the data of the test groups (compression and non-compression) to those of the controls in the respective assays by donors and analysis of mean, median, and variances be carried out using this new data in order to statistically assess whether compression increases the formation of nodules.
My second question is about the lineage used. Most studies describe the DU145 strain as being AR-negative, despite expressing levels of mRNA from the AR gene. However, many secondary bone tumors are AR-positive (DOI: 10.1677/ERC-10-0059) and the standard treatment involves the use of endocrine therapy. In this regard, there is a high netword interaction between testosterone/AR and IL-6 (PMID: 8988055; DOI: 10.1359/JBMR.050803), endocrine therapy impacts IL-6 levels (DOI:10.1002/cncr. 33398) and testosterone itself inhibits osteoclastgenesis (DOI: 10.1016/S0014-5793(01)02160-3), even though this phenomenon was not observed by the authors. Even if the CM from the compression groups, which led to an apparent higher production of nodules, contains lower levels of IL-6, the strain employed in relation to such mechanisms makes me question whether the results obtained could be generalized, or whether AR strains -positive would have different behavior. This should be discussed in the limitations.
Check lines 288-290. In the article of ref. 40 mentions the increase of osteoclastogenesis by GDF15, while that of ref.41 suppression by GDF15 and CM containing GDF15 from LNCaP.
Lines 306 to 310 attempt to relate mRNA expression to nodule formation. However, even though there was a decrease in the expression of the TGFB1 and TGFB2 genes, which would be contrary to the observed increased nodule formation, they are in a latent form and depend on enzymatic activation for release and activity. The justification presented by the authors is not appropriate and in line with the physiological process. Similarly, the translocation of beta-catenin, and not its expression at the mRNA or even protein level, is more relevant for processes such as EMT. Authors must address these issues.
Round 2
Reviewer 2 Report (New Reviewer)
I agree with the amendments and answers presented, except for one of paramount importance. The Friedman test is a non-parametric test for dependent samples, ie, repeated/paired measures/samples, while the Kruskal-Wallis test is for independent measures/samples. Therefore, they are not interchangeable. According to the reference presented by the authors (laerd.com), the Kruskal-Wallis teste is used to compare medians when distributions are similar ("similar variances") or mean ranks when they are not.
Robust versions of independent ANOVA for violation of homoscedasticity are by covariance matrix adjustment (HC3 and HC4 matrices mainly; for GLM) or, at least, Welch correction with Games-Howell post hoc. However, if there is a violation even of the normality of the distribution, or at least of symmetry (with adequate kurtosis), the Kruskal-Wallis test, or a GzLM, must be implemented. Therefore, the authors must make these methodological corrections.
Round 3
Reviewer 2 Report (New Reviewer)
I agree with the changes
This manuscript is a resubmission of an earlier submission. The following is a list of the peer review reports and author responses from that submission.
Round 1
Reviewer 1 Report
AUTHORS
Van Santen et al. investigated in vitro the effects induced by compressive stress on the behaviour of DU145 Prostate Pancer (PCa) cells. In particular, the Authors assesed whether the mechanical compression of DU145 PCa cells may modulate the expression of genes fosteting the epithelial-to-mesenchymal transition (EMT) and to the secretome of these cells as well. These studies hihlighted the fact that, the mechanical compression enhances the expression of some EMT-related genes and alters the secretome DU145 cells, decreasing osteoclast resorption activity and increasing osteoblast mineralization. These effects ultimately, may facilitate the cross-talk between PCa cells and bone marrow cells in order to rearrange the bone-tumor microenvironment to foster bone colonization by malignant cells
The subject of the paper appears to be timely and of potential interest to the IJMC readers. However, there are some key aspects of the submission that need to be addressed before it can be considered for publication
Introduction
1.Lines 50-52…”.these mechanical stimuli modulate the communication between PC cells, osteoblasts, and osteoclasts, to further rearrange the bone-tumor microenvironment and induce bone lesions…..”
Recent studies indicate that osteocytes appear to be actively involved in the colonization of bone tissue by cancer cells (see for instance Delgado-Calle J, Bellido T. Physiol Rev. 2022 Jan 1;102(1):379-410.. Maroni and Bendinelli. Cancers (Basel). 2020 Jul 6;12(7):1812; Wang, W et al. Oncogene 2019, 38, 4540; Cui YX et al. Anticancer Res 2016, 36:1193 and refs no. 18,21,22 of the MS). Moreover,t experimental observations have shown that mechanical stimulation is a potent regulator of osteocyte–cancer cell interactions in triggering metastatic cascade, for breast and prostate cancer(Verbruggen SW et al.. Cancers (Basel). 2021 Jun 10;13(12):2906. Therefore, it should be interesting to assess the effects of a CM from compressed cells and non-compressed PCa cells on osteocyte behaviour
2.Lines 76-81 “However, in the absence of compression, PC cells express a multitude of paracrine signals such as platelet-derived growth factor (PDGF) and IL6 that affect osteoblast and osteoclast differentiation and activity in vitro”. “It is therefore conceivable that compression modulates PC cell behavior, e.g. EMT, and their communication with osteoblasts and osteoclasts, to interfere with normal bone remodeling and contribute to the formation of bone lesions (???)”
The logical connection between these two clauses is unclear ( would itmeans that… as non compressed cells express several paracrine signalling therefore, it’s likely that compression may affects PCa cells ?)
3. Line 85 “To address our hypothesis, we compressed the PC cell line DU145”
Why authors used DU145 PCa cells (from brain metastasis) and not PC3 (from bone metastasis) for these studies?
86 “….then quantified gene expression related to EMT…” Which one in particular and why?
Materials and methods
Line 283 Subsequently, conditioned medium (CM) from compressed and non-compressed DU145.
A systematic analysis of the secretome of compressed and non-compressed cells may be performed in order to identify the molecules which may likely account for the phenomena shown in Fig.3 and Fig. 4° and the effectors which contribute to modulate the cross-talk between PCa cells and bone marrow cells in order to foster bone colonization by malignant cells
Line298 “qPCR was conducted to measure the expression of KI67, CYR61, Snail, VMN, and CDH1”
Why do the authors determined just EMT-related genes such as CYR61, Snail, VMN, and CDH1” ? Other genes such as TGF-β, Twist, Wntβ/catenin, HIF-1α and PPARϒ are known to play a key role in the EMT ↔MET transition and contribute to the formation of a permissive microenvironment, (i.e.“metastatic niche) which supports the homing, colonization and growth of cancer cells in the bone. In particular TGF-β has been shown to be very sensitive to mechanical stimuli. (see also your ref.41) Therefore the effects of mechanical compression on the expression levels of these gene should be to evaluate.
Regarding the role o PPARϒ See also comments Lines 201-204 results and discussion)
Line 330-331” mixed (ratio 1:1) with osteoclast medium”.
Lines 361 ” mixed (ratio 2:1) with osteoclast medium”.
Please provide more details about OC and OB media (obtained from…by? Composition?)
Results and Discussion
Lines 163-166 However, only the secretome of compressed DU145 cells impaired the resorptive activity of osteoclasts, while enhancing bone nodule production by osteogenic-differ-entiated hASCs from 4 out of 6 human donors
See comments line 283 Materials and Methods
Lines 201-204 “However, only the secretome of compressed PC cells reduced the resorptive activity of osteoclasts, while it enhanced bone nodule production by 4 out of 6 O-hASC donors,
compared to CM from non-compressed PC cells”
Accumulating evidence highlights the fact that bone marrow adipocytes promote bone metastasis formation in prostate cancer. The pro-metastatic effects of bone marrow adipocyte appear to be, in part , mediated perixosome proliferator–activated receptor γ, a transcriptional regulator of adipocyte differentiation and inhibitor of OB (Herroon MK et al. Oncotarget. 2013; 4:2108-2123; Ahmad I et al Proc. Natl. Acad. Sci. USA. 2016;113:8290–8295). PPAR-γ has been found expressed in both osteoblasts and adipocytes, as well as in MSCs, suggesting its crucial role in regulating adipocyte formation and osteoblast development. In particular, this factor has been reported to play a crucial role in bone development by fostering bone marrow adipogenesis and inhibiting osteoblastogenesis. Furthermore, PPARγ expression has been found in various ofcancer tissues and cell lines. Prostate cancers were found to overexpress PPARγ (Wagner N, Wagner KD. Cells. 2022 Aug 5;11(15):2432. doi: 10.3390/cells11152432.) . In metastatic prostate cancer,. PPARγ promotes the growth of this cancer type via the activation of lipid signaling pathways, (Liu R.Z., Mol. Oncol. 2020;14:3100–3120.) . In ths context PPARγ has been reported to accelerate EMT transition (Galbraith. Oncogene. 2021;40:2355–2366. doi: 10.1038/s41388-021-01707-7. )
Therefore it would be interesting to investigate the effects of mechanical compression on the expression of PPARγ.
Reviewer 2 Report
Mechanical stresses significantly alter cancer cell properties and influence cancer progression, metastasis, and treatment response. In this study, Santen et al. briefly investigated the DU145 prostate cancer cell line in the context of compression with in vitro models. They concluded that compressing DU145 cells changes their DNA/RNA expression and secretome. It is an exciting topic, unfortunately, as the authors claimed, this study has quite some limitations. In addition to these pointed out by the authors, here are more concerns that need to be addressed.
1. This is a descriptive study that is performed with a single cell line, no mechanism has been proposed and tested.
2. The authors used 2 kPa compression. Although the compression stabilized at 2 kPa 20 days after colonizing to the bone, the change from 5 kPa to 2 kPa may also significantly affect cancer cell properties. The use of 2 kPa compression with 2 days needs to be justified.
3. A few EMT-related genes were altered upon compression. As many genes are involved in EMT, the changing of these indicated genes does not guarantee EMT. The authors should examine if the EMT phenotype can be observed.
4. The figure legends need to be described in detail. For instance, in Figure 1, I do not understand how total DNA and RNA per cell are evaluated.
Reviewer 3 Report
This study is interesting in that it examines the interaction of prostate cancer cells with their environment after they have metastasized to bone tissue from a stress perspective. It is informative for studying the treatment of prostate cancer after bone metastasis.
I would like to suggest some minor changes for the author to revise the manuscript.
1. It would be more space efficient to have one or two lines for Figures 1 and 2.
2. In Figure 3, the scales for A and B need to be updated, as the current scales are not clearly visible.
3. In Figure 4, a scale needs to be added to A.
4. To enhance the readability of the article, I suggest that the role of the snail, CYR61, VMN and CDH1 genes in the EMT process be presented in section 2.2 of the results so that readers not in this field can better understand the study.
Round 2
Reviewer 2 Report
Thanks for providing more discussion and clarification.